# Influence of Ascorbic Acid as a Growth and Differentiation Factor on Dental Stem Cells Used in Regenerative Endodontic Therapies

**DOI:** 10.3390/jcm12031196

**Published:** 2023-02-02

**Authors:** Antje Diederich, Hanna Juliane Fründ, Bogusz Trojanowicz, Alexander Navarrete Santos, Anh Duc Nguyen, Cuong Hoang-Vu, Christian Ralf Gernhardt

**Affiliations:** 1University Outpatient Clinic for Conservative Dentistry and Periodontology, Martin-Luther-University Halle-Wittenberg, 06112 Halle, Germany; 2Department of Visceral, Vascular and Endocrine Surgery, University Medical Center Halle, Martin-Luther-University Halle-Wittenberg, 06120 Halle, Germany; 3Center for Medical Basic Research, Martin-Luther-University Halle-Wittenberg, 06108 Halle, Germany; 4Private Dental Practice, Dr. Juliane Gernhardt, 06120 Halle, Germany

**Keywords:** ascorbic acid, dental pulp stem cells, regenerative endodontic procedures, stem cell markers

## Abstract

Background: Vitamin C is one of the major extracellular nonenzymatic antioxidants involved in the biosynthesis of collagen. It promotes the growth of fibroblasts, wound healing processes, and enhances the survival and differentiation of osteoblasts. The potential effects of ascorbic acid on human dental pulp cells (DPC) and the cells of the apical papilla (CAP) used in actual regenerative endodontic procedures remain largely unknown. In this study, we investigated the possible employment of ascorbic acid in the differentiation and regenerative therapies of DPC and CAP. Methods: Nine extracted human wisdom teeth were selected for this study. Subpopulations of stem cells within DPC and CAP were sorted with the mesenchymal stem cell marker STRO-1, followed by treatments with different concentrations (0 mM, 0.1 mM, 0.5 mM, and 1.0 mM) of ascorbic acid (AA), RT-PCR, and Western blot analysis. Results: FACS analysis revealed the presence of cell subpopulations characterized by a strong expression of mesenchymal stem cell marker STRO-1 and dental stem cell markers CD105, CD44, CD146, CD90, and CD29. Treatment of the cells with defined amounts of AA revealed a markedly increased expression of proliferation marker Ki-67, especially in the concentration range between 0.1 mM and 0.5 mM. Further investigations demonstrated that treatment with AA led to significantly increased expression of common stem cell markers OCT4, Nanog, and Sox2. The most potent proliferative and expressional effects of AA were observed in the concentration of 0.1 mM. Conclusions: AA might be a novel and potent growth promoter of human dental cells. Increasing the properties of human dental pulp cells and the cells of the apical papilla using AA could be a useful factor for further clinical developments of regenerative endodontic procedures.

## 1. Introduction

Dental tissue engineering and the regeneration of functional tooth–tissue structures are actually interesting and remarkable topics in endodontics [1,2]. This area involves multidisciplinary approaches based on the interaction of three basic key elements: stem cells, morphogens, and scaffolds [3]. In this connection, locally niched stem cells and their differentiation and pluripotency are of considerable importance, together with the morphogens, growth factors, and cytokines that are intrinsically involved in the process of odontogenesis. This is an important factor because, following this idea, it is able to produce dentin, which maintains the biological and physiological vitality of the tooth. Due to the fact that dental caries is still one of the most prevalent health problems in dentistry [4] and a major cause for the early loss of dental pulp vitality and subsequent tooth deficiency, the need for successful pulp regeneration therapies in endodontics is increasing [5]. The modulation of dental stem cells is one topic in this field. Ascorbic acid (AA), also called vitamin C, is an important antioxidant and plays a crucial role in many biological systems. It is the major nonenzymatic, water-soluble antioxidant and a regulator of bone cells metabolism and vitality. It is also known to reduce apoptosis [6,7]. In connective tissues, AA works as a cofactor of lysine and proline hydroxylase involved in collagen synthesis [8]. Reports from in vitro studies demonstrated that AA is able to decrease the activity and survival rates of osteoclasts while promoting the vitality and differentiation of osteoblasts [9,10,11]. In cooperative actions with vitamin A and E, AA is able to accelerate bone healing by promoting osteogenic activity [12]. With regard to dental-related disorders, it is well known that AA deficiency promotes scurvy, a disease of the periodontium [13]. However, the potential effects of AA on human dental pulp cells (DPC) or cells of the apical papilla (CAP) remain unknown. Recently, the rising interest in employing human dental cells in replacement therapies, especially DPC, presents these cells as an interesting and efficient candidate for regenerative endodontic procedures. Moreover dental follicle stem cells (DFSC), stem cells of the apical papilla (SCAP), and periodontal ligament cells (PDL) [14], DPCs contain a population of dental pulp stem cells (DPSC), which are able to differentiate into other cell types such as osteoblasts-, adipocytes-, and neuronal-like cells, and the serrate mineralized matrix [15,16]. Based on our previous observations that DPSCs contain a wide spectrum of pluripotency markers [17,18], in this study, we investigated the potential influence of AA on DPC and CAP. Furthermore, the optimal concentration of AA and AA-mediated effects on these cells and their possible employment in prospective clinical regenerative endodontic therapies were tested. The hypothesis to be tested was that AA has a positive effect on the growth and proliferation of different dental cell types.

## 2. Materials and Methods

### 2.1. Isolation of Primary Cells

Human DPC and CAP were isolated as described by Gronthos et al., 2000 [19]. The cells were extracted from wisdom teeth of young patients between 16 and 25 years old, removed due to the medical indication of dental surgery at the dental clinic Ernst & Schenk, Halle (Saale). Within one hour, the apical papilla was separated under sterile conditions and the tooth was gently opened with dental instruments in order to reach the pulp cells. The pulp and the apical papilla were dissociated with a scalpel. Thereafter, the tissues were digested by employing an enzyme mixture containing 3 mg/mL of collagenase type I and 4 mg/mL of dispase in Hank’s Balanced Salt Solution (HBSS from Gibco, Darmstadt, Germany) at 37 °C for 90 min with shaking. Afterwards, the cells were further dissociated from each other by employing a cell strainer with a pore size of 70 μm. The cells were cultured in growth medium (α-MEM, Gibco, Darmstadt, Germany) supplemented with 100 nM of L-ascorbic-acid-2-phosphate (Sigma-Aldrich Chemie, Steinheim, Germany); 2 mM of L-glutamine (Merck, Darmstadt, Germany); PenStrep (Gibco, Darmstadt, Germany); Amphotericin B (Fungizone^®^, Gibco, Darmstadt, Germany); and 10–15% fetal calf serum (FCS, Gibco, Darmstadt, Germany). The cells were passaged at 8% confluence. After isolation of the cells from the teeth, the sorting of the cells was performed using FACS in the flow cytometry procedure. The cells were stained with STRO-1 markers and then divided into STRO-1-positive and STRO-1-negative cells based on their autofluorescence. Treatment of the cells was performed at indicated times (24 h, 48 h, and 72 h) and with specified concentrations (0 mM, 0.1 mM, 0.5 mM, and 1 mM) of AA (Carl Roth, Karlsruhe, Germany). The pH was measured digitally at different times at 37 °C. To examine RNA expressions, RNA was isolated from the cells and assayed in a quantitative real-time PCR. Positive controls were then used to quantify these. For the analysis of specific protein expressions, two methods were performed in this study: first, the immunocytochemical staining of cells with selected antibodies and, second, the isolation of proteins and their detection using a Western blot. For each concentration, 18 cell cultures were inoculated; one culture was not treated with AA and was used as a control group.

### 2.2. FACS Analysis

Subpopulations of DPSC and SCAP contained within DPC and CAP, respectively, were sorted according to their mesenchymal stem cell marker STRO-1 expression by employing mouse IgM antihuman STRO-1 (Anti-STRO-1 IgM, R & D Systems Inc., Minneapolis, MN, USA, 10 μg/mL) and PE-conjugated secondary antibodies, respectively (1:50, Sigma–Aldrich Chemie GmbH, Steinheim, Germany). Both antibodies were incubated for 30 min on ice and washed with PBS. Dental cells were sorted with twofold flow cytometric cell sorting (FACS Aria II, BD Biosciences, Franklin Lake, NJ, USA).

### 2.3. Real-Time PCR

Total RNA was isolated from 3.5 × 10^4^ cells/mL by employing TRIzol^TM^ reagent (Invitrogen™, Carlsbad, CA, USA). An amount of 1 μg of total RNA was used as a template for first-strand cDNA synthesis (SuperScript™ II Reverse Transcriptase, Invitrogen™, Carlsbad, CA, USA) according to the manufacturer’s instructions. The samples were stored at 20 °C. Amplifications of OCT4, Ki-67, CD105, CD44, CD146, CD90, CD29, and GAPDH were performed with the Rotor-Gene Q Realtime qPCR System (Qiagen GmbH, Hilden, Germany) and SYBR Green MasterMix according to the manufacturer’s instructions (Qiagen GmbH, Hilden, Germany). The following primer pairs were used: OCT4 sense: 5′-GAGAAGGAGAAGCTGGAGCA-3′, antisense: 3′-AATAGAACCCCCAGGGTGAG-5′; GAPDH sense: 5′-ACCCAGAAGACTGTGGATGG-3′, antisense: 3′-TTCTAGACGGCAGGTCAGGT-5′; Ki-67 sense: 5′-GTGGGCACCTAAGACCTGAA-3′, antisense: 3′-ATGGTTGAGGCTGTTCCTTG-5′; CD29 sense: 5′-AACTGCACCAGCCCATTT-3′, CD29 antisense: 3′-AGCCAATCAGTGATCCACAA-5′; CD44 sense: 5′-GACCTCTGCAAGGCTTTCAAT-3′, CD44 antisense: 3′-AATCACCACGTGCCCTTCTAT-5′; CD90 sense: 5′-TCCCGAACCAACTTCACCAG-3′, CD90 antisense: 3′-GATGCCCTCACACTTGACCA-5′; CD105 sense: 5′-ACTCTCCAGGCATCCAAGC-3′, CD105 antisense: 3′-GGAAGGATGGCAGCTCTGT-5′; and CD146 sense: 5′-GTCGTCCCAGACTGTGGAGT-3′, CD146 antisense: 3′-CCACTTCCAGCCACACTTTT-5′. SW480 (cancer cell line of the colon) with a high expression of stem cell markers was used as a positive control in all qPCR reactions. All amplified target genes were normalized with GAPDH, compared to positive control, set as 100%.

### 2.4. Immunohistochemistry

The dental cell lines DPC and CAP were seeded on Thermanox coverslips™ (Invitrogen™, Carlsbad, CA, USA) in 24-well plates and grown in α-MEM. After stimulation with the corresponding AA concentrations, the membranes were washed with PBS and fixed in a 1:4 mixture of 3% H_2_O_2_ and 90% methanol for 20 min. The membranes were washed with PBS again. Primary antibodies directed against FasR (1:500) and FasL (1:20) were diluted in Dako Antibody Diluent (Dako GmbH, Hamburg, Germany) and incubated overnight. Thereafter, Thermanox membranes were first incubated with Dako bridge antibody and, after that, with Dako streptavidin-peroxidase, each for 30 min. Antibody staining was performed with Dako substrate reagent. Microscopic investigations were performed with a conventional light microscope (Carl Zeiss, Jena, Germany).

### 2.5. Western Blot

Total proteins were isolated with lysis buffer (7 M Urea, 2 M Thiourea, 4% CHAPS, 40 mM DTT, 0.5% Pharmalyte 3–10) supplemented with a protease inhibitor cocktail (all reagents were from Sigma-Aldrich GmbH, Steinheim, Germany). An amount of 10 μg of proteins was resolved on a 10% SDS-PAGE and subsequently transferred onto PVDF membrane (GE Healthcare, Chicago, IL, USA). After blocking with 5% BSA/TBST for 1 h, the membranes were incubated with primary antibodies (OCT-4 1:1000, Nanog 1:500, CD105, Merck KGaA, Darmstadt, Germany) overnight. After incubation, the membranes were washed with TBS-T three times, 10 min each, and again incubated with horseradish peroxidase (HRP)-conjugated goat anti-rabbit IgG (1:20,000, Santa Cruz Biotechnology, Dallas, TX, USA) for 1 h at room temperature. After washing with TBS-T three times, the immunoreactive protein bands were visualized with the ECL kit (Merck KGaA, Darmstadt, Germany). Human β-actin served as the normalizing marker and total lysates from cell line PATU-8988S (pancreatic carcinoma) was used as positive control as cancerogenic cell lines express increasing stem cell markers.

### 2.6. Statistical Analysis

Each experiment was repeated at least three times. Data are presented as mean ± SD. Distribution of the quantitative variables was tested using D’Agostino–Pearson omnibus normality test. Depending on data distribution, parametric (differences between paired values are consistent) or nonparametric (Wilcoxon matched-pairs signed rank test) two-sided *t* tests were used. *p* < 0.05 was considered to represent statistically significant differences. Prism 9 software (GraphPad, Boston, MA, USA) was used for statistical analysis.

## 3. Results

### 3.1. STRO-1 Correlates with Increased Expression of Stem Cell Markers in DPSCs and CAPs

CAP and DPC revealed a fibroblast-like morphology. In the first passages, the CAP showed a slower growth than DPC. However, in later passages, there were no considerable differences visible. In further investigations, the cells were subjected to FACS analysis in order to estimate the number of the stem cells positive for STRO-1. As demonstrated in Figure 1, the rate of STRO-1-positive cells was 1.5% for DPCs (Figure 1A), whereas for CAPs, it was 4.5% (Figure 1B). In order to investigate whether STRO-1-positive DPCs (DPSCs) or CAPs (SCAPs) may possess an increased regenerative potential, we subjected these cells to RT-PCR with primers specific for common stem cell markers. As shown in Figure 1C–G, the cells positive for STRO-1 revealed significantly increased expression of CD29, CD44, CD90, CD105, and CD146 as compared to the unsorted population. Elevated expression of these markers indicates that STRO-1-positive cells (DPSC and SCAP) may retain similar stem cell properties and potential like other mesenchymal stem cells.

### 3.2. AA Promotes the Proliferation of DPC and CAP

Current reports suggest that the optimal growth-promoting concentration of AA for DPC and CAP is unknown [20]. To address this question, we subjected the cells to different AA concentrations and investigated their proliferative properties with an analysis of Ki-67 expression. Additionally, to exclude the fact that the observed effects were not due to the pH alteration, treatment media with different AA concentrations were investigated for pH variations. As shown in Figure 2E, pH was constant with all the AA concentrations tested. As demonstrated in Figure 2A–D, the AA concentration of 0.1 mM led to the highest induction of proliferative marker Ki-67 as compared to the other concentrations tested. These effects were observed for total (DPC and CAP) and sorted cell subpopulations (DPSC and SCAP). Based on these observations, we chose this concentration in further experiments. Furthermore, the expression of apoptosis marker Fas remained unaffected under treatment with different concentrations of AA. FasL was not detected in any AA concentrations tested (data not shown). These data suggest that AA-mediated effects were, under the tested conditions, pH- and apoptosis-independent.

### 3.3. AA Induces Stem Cell Markers in DPC and CAP

In order to estimate the potential effects of AA on the expression of common stem cell markers, we subjected unsorted DPC and CAP, and STRO-1-positive DPSC and SCAP, to treatment with 0.1 mM AA. RT-PCR analysis revealed that AA led to the significant upregulation of OCT4, Nanog, and Sox2 as compared to corresponding controls (Figure 3A–D). Similar results demonstrating the significant elevation of the stem cell markers OCT-4, Nanog, and CD105 were also obtained on the protein level (Figure 4A–D).

## 4. Discussion

In the present study, we demonstrated that AA promoted the proliferative and regenerative abilities of human DPC and CAP. Through the employment of transcript and protein analysis, we found that 0.1 mM AA was the optimal proliferative concentration for the upregulation of stem cell markers crucial for cellular pluripotency.

The optimal concentration of AA for regenerative dental therapies is currently unknown. The renewal of functional structures of dental tissues is a multistep process dependent on the coordinated interaction between stem cells, morphogens, and scaffolds [3,21]. The influence of growth factors, cytokines, and, especially, morphogens on the local stem cells is crucial for the processes of odontogenesis [22]. We found that AA, a potent morphogen regulating cellular proliferation and the differentiation of osteoblasts, adipocytes, chondrocytes, and odontoblasts [23,24,25,26,27,28], also exerts the proliferative effects on DPC and CAP. Previous reports demonstrated the positive effects of AA on the differentiation of stem cells, such as human bone-marrow-derived mesenchymal stem cells or mouse embryonic stem cells [29,30].

Phenotypically, dental stem cells express a range of surface markers including CD44, STRO-1, CD90, CD105, CD106, and CD146. It is also worth noting that mesenchymal stem cells are a rich source of these antigens [31,32,33]. In accordance with previous reports, we found that our cells not only express these markers, but also the markers of pluripotency, such as OCT-4, Nanog, and Sox2. Furthermore, the most important finding of this study, the employment of AA in DPC or CAP or STRO-1-positive subpopulations of these cells led to the enhanced expression of this broad spectrum of different pluripotency-related markers. We observed that AA-mediated effects affected not only the expression of dental stem cell markers, but also the levels of Ki-67, a mitotic protein commonly used as a marker for proliferating cells [10].

The employment of AA in DPC and CAP led to the upregulation of Ki-67. Furthermore, the effects of AA were not related to apoptosis induction as the expressions of Fas or FasL were not changed by treatment with 0.1 mM or 0.5 mM of AA. However, as described previously [7], a higher concentration of AA may inhibit Fas-induced apoptosis and the Fas/FasL system may apparently be involved in tooth development [34,35]. In this study, 1 mM of AA had no significant influence on Fas expression. Although our hypothesis could be confirmed, the limitations of the present study might be located in the used primary cell cultures. Furthermore, while in vitro culture systems offer a widely used model to investigate stem cell growth and differentiation, they fall short in their ability to allow the exact determination of the stem cells’ ability to generate functional tissues including dentin-imitating architecture. Therefore, further investigations using in vivo transplantation systems, the way towards clinical implementation, are needed to improve clinical regenerative endodontic procedures.

## 5. Conclusions

The present study reveals that DPC and CAP, and STRO-1-positive stem cell subpopulations originating from both these types, respond to 0.1 mM AA treatment with increased expression of dental stem cell markers and enhanced proliferation ability. AA in the concentration of 0.1 mM is a potent growth inductor with high potential in prospective scientifically based endodontic regenerative therapies. These findings justify further investigation and verification of these findings, first in animal models and, later on, in clinical settings. AA is a novel and potent growth promoter of human dental cells. By increasing the properties of human dental pulp cells and the cells of the apical papilla, it might be helpful to improve the outcome of regenerative endodontic procedures in our patients.

## Figures and Tables

**Figure 1 jcm-12-01196-f001:**
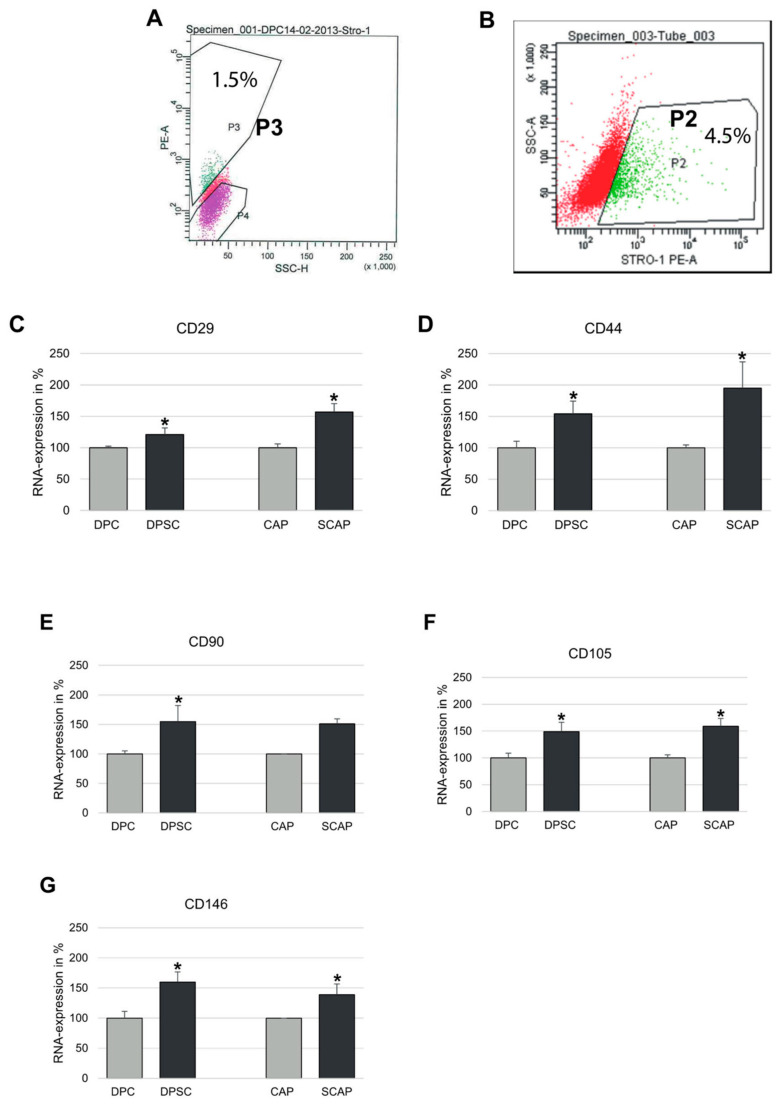
FACS-mediated sorting and expression of mesenchymal stem cell markers in DPC and CAP. (**A**) DPC and (**B**) CAP cell populations were sorted according to the expression of stem cell marker STRO-1; STRO-1-positive subpopulations were named as DPSC and SCAP, respectively; representative gates P3 (1.5% of DPSC) and P2 (4.5% of SCAP) demonstrate PE (phycoerythrine) STRO-1-positive cells. (**C**–**G**) Expression of mesenchymal stem cell markers in STRO-1-negative (DPC and CAP) and STRO-1-positive (DPSC and SCAP) DPC and CAP. All cell populations were tested for the expression of (**A**) CD29, (**B**) CD44, (**C**) CD90, (**D**) CD105, and (**E**) CD146; means ± SD of three independent experiments; * *p* < 0.05 indicates statistical significance (vs. STRO-1-negative cells).

**Figure 2 jcm-12-01196-f002:**
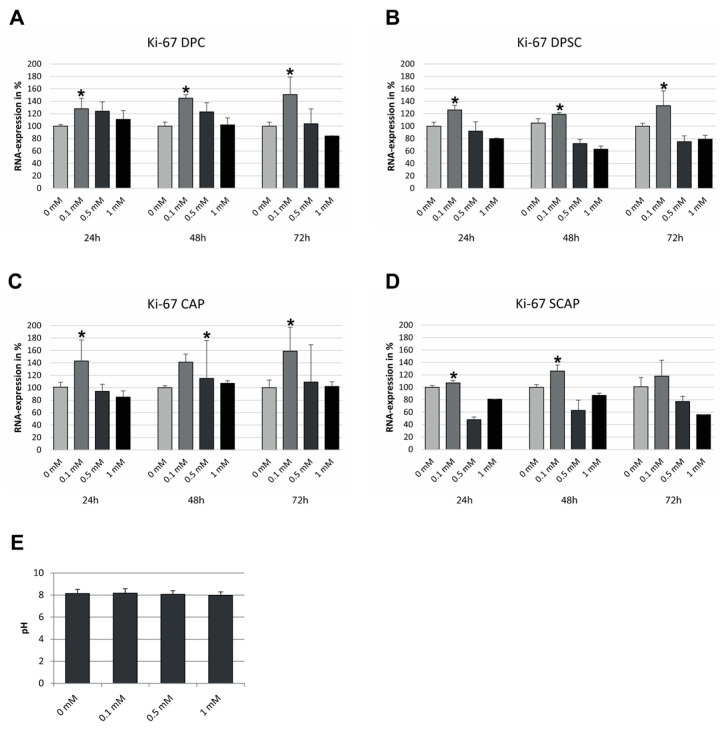
Proliferation of the cells after treatment with different concentrations of AA, based on the expression of Ki-67. DPC (**A**), (**B**) DPSC, (**C**) CAP, and (**D**) SCAP were treated with 0.1 mM, 0.5 mM, and 1 mM AA for 24 h, 48 h, and 72 h, and subjected for Ki-67 analyses. (**E**) Measurement of pH with indicated AA concentrations. Means ± SD of three independent experiments; * *p* < 0.05 indicates statistical significance vs. 0 mM.

**Figure 3 jcm-12-01196-f003:**
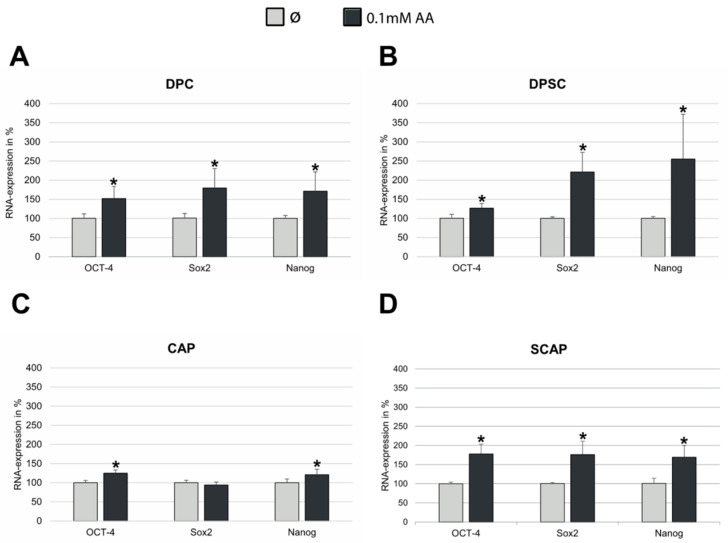
Expression of stem cell markers OCT-4, Sox2, and Nanog in (**A**) DPC, (**B**) CAP, (**C**) DPSC, and (**D**) SCAP treated with 0.1 mM AA. Means ± SD of three independent experiments; * *p* < 0.05 indicates statistical significance vs. Ø (0 mM AA).

**Figure 4 jcm-12-01196-f004:**
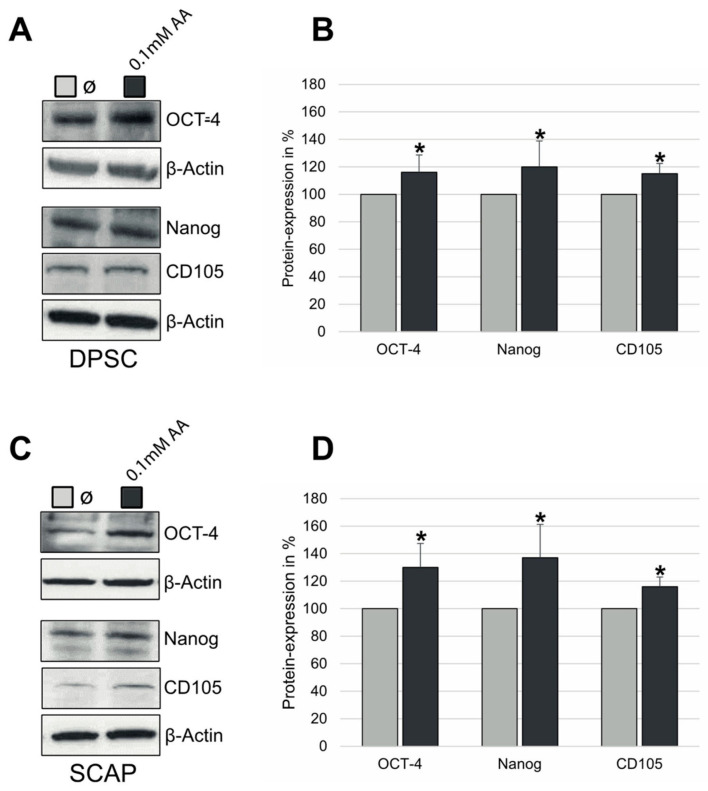
Protein expression of OCT4, Nanog, and CD105 in (**A**,**B**) DPSC and SCAP (**C**,**D**) treated with 0.1 mM AA. Means ± SD of three independent experiments; * *p* < 0.05 indicates statistical significance vs. Ø (0 mM AA).

## Data Availability

The data are available upon request from the corresponding author.

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
