# Peer review of "Influence of Ascorbic Acid as a Growth and Differentiation Factor on Dental Stem Cells Used in Regenerative Endodontic Therapies"

_jcm, 2023, doi:10.3390/jcm12031196_

Round 1

Reviewer 1 Report

Influence of Ascorbic Acid as a Growth and Differentiation Factor on Dental Stem Cells Used in Regenerative Endodontic Therapies

    This article is original and interesting; however, revision is suggested. 

Abstract: Please to provide brief conclusion.

1.     Introduction:

1.1  Please to explain the rationale of “Ascorbic Acid as a Growth and Differentiation

Factor on Dental Stem Cells” clearly. 

1.2. Lack of hypothesis and the aim of this study, please to add the information.

2.     Materials & Methods:

2.1  The study was approved by the Ethics Committee of the 247 Medical Faculty of the Martin-Luther-University Halle-Wittenberg on 13.01.2010.                               à Does it mean that IRB approved in 2010?                                                                  Was the article finished in 2022? When these wisdom teeth (sample) were extracted? And, please to provide IRB approval number.

2.2  FACS analysis: Please reconfirm “Clone STRO-1, R&D Systems, 10μg/ml” and “FACS Vantage, BD Bioscience” whether lack of information.

2.3  Real-time PCR: Please reconfirm “Trizol reagent” and “SuperScriptTM II 97 KIT, Life Technologies” whether lack of information.                                           Why cell line SW480 was used as a positive control? 

2.4  Immunohistochemistry: Please reconfirm “Thermanox membranes” and “Dako Antibody Diluent” whether lack of information.

2.5  Western Blot: Why cell line PATU-8988S was used as a positive control?

3.     Results:

3.1  AA promotes the proliferation of DPC and CAP:                                            Current reports suggest that the optimal, growth-promoting concentration of AA for DPC and CAP is unknown. Please to provide citation.                                                                       How to measure pH? Did the test method miss in “Materials & Methods”?

4.     Discussion:

4.1  The influence of growth factors, cytokines and especially morphogens on the local stem cells is crucial for the processes of odontogenesis. à please to provide citation.

4.2  In this study, 1mM AA had no significant influence on Fas expression. à Did it mean that 1mM AA had influence on Fas expression which was not significant?

4.3  Please to discuss the finding in Figure 2D(Ki-67 SCAP) because it was different from findings in Figure 2A-2C.

4.4  Please to provide the limitation of this study and further study direction.

5.     Conclusion: Lack of conclusion, please to provide summary from the results.

6.     Figures:                                                                     

6.1  Please reconfirm star mark (*) in Figure 2C.

6.2  Why pH of “0.1mM AA” concentration was higher than that of “0.1mM AA” in Figure 2E?

6.3  What is “Ø” in Figure 3 and Figure 4?

Author Response

Response to Reviewer 1

The point-by-point response to the reviewer’s comments including all points raised by reviewer 1. Thank you very much for reviewing our paper and all the helpful comments.  

Influence of Ascorbic Acid as a Growth and Differentiation Factor on Dental Stem Cells Used in Regenerative Endodontic Therapies. This article is original and interesting; however, revision is suggested. 

 Abstract: Please to provide brief conclusion.

 A conclusion was included in the abstract.

  1. Introduction:

1.1  Please to explain the rationale of “Ascorbic Acid as a growth and differentiation factor on dental stem cells” clearly. 

See improved introduction section. This point was clarified.

1.2. Lack of hypothesis and the aim of this study, please to add the information.

See improved introduction section. This point was added.

  1. Materials & Methods:

2.1  The study was approved by the Ethics Committee of the 247 Medical Faculty of the Martin-Luther-University Halle-Wittenberg on 13.01.2010. Does it mean that IRB approved in 2010? Was the article finished in 2022? When these wisdom teeth (sample) were extracted? And, please to provide IRB approval number.

The study is part of a long-term scientific project called “Tissue Engineering of Teeth”. This project was approved by the ethics committee of the Martin-Luther-University Halle-Wittenberg on 13.01.2010 before starting the first part of the project. A copy of the original approval is included. This part of the project was carried out 2020-2022. The wisdom teeth used in the study were freshly extracted and stored in saline for a maximum of 1 hour after extraction.

2.2  FACS analysis: Please reconfirm “Clone STRO-1, R&D Systems, 10μg/ml” and “FACS Vantage, BD Bioscience” whether lack of information.

The missing information is included in the manuscript.

2.3  Real-time PCR: Please reconfirm “Trizol reagent” and “SuperScriptTM II 97 KIT, Life Technologies” whether lack of information. Why cell line SW480 was used as a positive control? 

The missing information is included in the manuscript. The explanation for cell line SW480 was also included.

2.4  Immunohistochemistry: Please reconfirm “Thermanox membranes” and “Dako Antibody Diluent” whether lack of information.

The missing information is included in the manuscript.

2.5  Western Blot: Why cell line PATU-8988S was used as a positive control?

The missing information is included in the manuscript. The explanation for cell line PATU-8988S as positive control was also included.

  1. Results:

3.1  AA promotes the proliferation of DPC and CAP: Current reports suggest that the optimal, growth-promoting concentration of AA for DPC and CAP is unknown. Please to provide citation. How to measure pH? Did the test method miss in “Materials & Methods”?

Please also see the discussion section. Citation was included in 3.2. The ph was measured using a digital ph-meter. However, the use of ascorbic acid in a buffered solution (α-mem) does not has any impact on ph.

  1. Discussion:

4.1  The influence of growth factors, cytokines and especially morphogens on the local stem cells is crucial for the processes of odontogenesis. à please to provide citation.

The missing citation is now included in the manuscript.

4.2  In this study, 1mM AA had no significant influence on Fas expression. à Did it mean that 1mM AA had influence on Fas expression which was not significant?

Yes, it means that 1mM AA had influence on Fas expression which was not significant.

4.3  Please to discuss the finding in Figure 2D(Ki-67 SCAP) because it was different from findings in Figure 2A-2C.

A small scatter of the values is dependent on the temperature and time of the measurement.

4.4  Please to provide the limitation of this study and further study direction.

  1. Conclusion: Lack of conclusion, please to provide summary from the results.

The conclusion has now been completed.

  1. Figures:                                                                     

6.1  Please reconfirm star mark (*) in Figure 2C.

Thank you very much for this correction. Therefore, the star mark is now corrected in Figure 2C.

6.2  Why pH of “0.1mM AA” concentration was higher than that of “0.1mM AA” in Figure 2E?

The response to AA is increased at 0.1 for all cell types studied. At higher concentrations the cell types respond differently.

6.3  What is “Ø” in Figure 3 and Figure 4?

This symbol is now explained in the chart legend.

Thank you.

Reviewer 2 Report

very good manuscript. I would just ask about the ethics committee. Provide a copy of the committee

Author Response

Response to Reviewer 2

The point-by-point response to the reviewer’s comments including all points raised by reviewer 2. Thank you very much for reviewing our paper.

I would just ask about the ethics committee. Provide a copy of the committee.

The study is part of a long-term scientific project called “Tissue Engineering of Teeth”. This project was approved by the ethics committee of the Martin-Luther-University Halle-Wittenberg on 13.01.2010 before starting the first part of the project. Due to the interdisciplinary approach of the project, the original approval is located at the ethics committee of the Martin-Luther-University Halle-Wittenberg. If necessary, we can ask for a copy.

Thank you.

Reviewer 3 Report

I was pleased to review the article ijerph-2177903 entitled “Influence of Ascorbic Acid as a Growth and Differentiation Factor on Dental Stem Cells Used in Regenerative Endodontic Therapies” for the Journal of Clinical Medicine.

Overall, the article does not look like it fits the Journal’s scope.

Moreover,

The introduction needs to be enriched.

It needs to be clarified in the introduction what the scientific gap the systematic review addresses.

What is the relevance of the manuscript’s findings?

How the results can be used for the development of new biomaterials.

The inclusion of osteogenic markers is suitable.

Author Response

Response to Reviewer 3

The point-by-point response to the reviewer’s comments including all points raised by reviewer 3. First of all, thank you for reviewing our paper and the helpful comments.

Overall, the article does not look like it fits the Journal’s scope.

In accordance and after discussion with both special issue editors, Prof. Schwendicke and Dr. Herbst, we decided to submit the article as it focuses on an interesting point in clinical endodontics- the regenerative approach in endodontic therapy. The findings of our study might help to improve regenerative procedures in endodontics. So, together with the editors we think it fits in the journal´s scope.

Moreover, the introduction needs to be enriched. It needs to be clarified in the introduction what the scientific gap the systematic review addresses.

The introduction section was improved. Please see the revised manuscript.

What is the relevance of the manuscript’s findings?

A conclusion section was added and the relevant findings were included.

How the results can be used for the development of new biomaterials.

This point was also addressed and included in the manuscript. Our findings might help to improve the clinical procedure not the materials used.

The inclusion of osteogenic markers is suitable.

Thank you.

Round 2

Reviewer 1 Report

none

Reviewer 3 Report

The authors improved the manuscript quality. It is suitable to be accepted in its current version.